# Implementation of Virtual Reality in Preclinical Pediatric Dentistry Learning: A Comparison Between Simodont^®^ and Conventional Methods

**DOI:** 10.3390/dj13020051

**Published:** 2025-01-23

**Authors:** Antonia M. Caleya, Andrea Martín-Vacas, María Rosa Mourelle-Martínez, Manuel Joaquín de Nova-Garcia, Nuria E. Gallardo-López

**Affiliations:** 1Department of Dental Clinical Specialties, Faculty of Dentistry, Complutense University of Madrid, 28040 Madrid, Spain; amcaleya@ucm.es (A.M.C.); mrmourel@ucm.es (M.R.M.-M.); denova@ucm.es (M.J.d.N.-G.); 2Faculty of Dentistry, Alfonso X El Sabio University, 28691 Villanueva de la Cañada, Spain; amartvac@uax.es

**Keywords:** pediatric dentistry, dental education, virtual reality

## Abstract

**Background/Objectives:** Preclinical training in pediatric dentistry is traditionally performed on acrylic primary teeth or natural extracted teeth in phantoms or dental manikins. With technological advancements, virtual simulation has become increasingly common, enhancing the development of cognitive and motor skills in dental students while complementing traditional methods. Specific objectives included assessing whether preclinical theoretical knowledge impacts motor skill scores, comparing the quality of dental preparations on acrylic teeth versus those performed using virtual simulation (Simodont^®^) and evaluating students’ perceptions of Simodont^®^ use. **Methods:** An observational, descriptive, cross-sectional study was conducted. Students first completed a theoretical knowledge survey on dental preparations, class II caries, and pulpotomies. They then performed dental preparations on both acrylic primary teeth and the Simodont^®^ simulator, with their work assessed by a pediatric dentistry professor. Finally, students completed a perception survey regarding their experience. **Results:** The introduction of the Simodont^®^ was positively received. Theoretical knowledge prior to preclinical exercises did not significantly influence practice scores. Average scores for preparations on acrylic teeth (class II: 2.57; pulpotomy: 3.60) were higher than those achieved using Simodont^®^ (class II: 1.97; pulpotomy: 2.92). **Conclusions**: Dental preparation scores were higher on acrylic teeth compared to the virtual simulation. While students reported a positive experience with Simodont^®^, they expressed a preference for traditional learning methods involving acrylic teeth on phantoms or dental manikins.

## 1. Introduction

Simulation scenarios are essential in dental education, as they provide students with a controlled and safe environment to develop their skills prior to treating real patients [1]. The training of dental students requires not only the acquisition of theoretical knowledge but also the development of motor skills necessary for clinical practice. Before engaging in patient care, it is crucial that students first attain the appropriate competencies and skills, which can be achieved through simulated preclinical exercises [2,3,4,5]. Preclinical practice plays a pivotal role in the development of clinical skills, offering a platform for student training as well as for the evaluation of their abilities and proficiency [6,7].

Simulated learning environments in dentistry enable students to enhance their clinical skills through a variety of structured activities [8]. Traditionally, mannequin heads equipped with natural cadaver teeth were used; however, these are used less because permits are required, since it has to be approved by the Ethics Committee. This shift necessitated the use of artificial acrylic teeth in phantoms [1]. While these conventional methods are well-established, they have notable drawbacks, including high material consumption, the limited availability of qualified personnel for supervision, and substantial initial costs. Moreover, the tactile feedback differs significantly from that of natural teeth, and the artificial teeth are single-use, restricting opportunities for iterative feedback. Additionally, acrylic primary teeth for pulpotomies are expensive and poorly replicate the properties of natural primary teeth [1,3,9].

With the rapid technological advancements in recent years, virtual simulators (VSs) have been introduced, supported by extensive evidence in both medical and dental education [10,11,12,13]. These innovations have significantly enhanced the cognitive and motor skill development of students. Haptic dental simulators, which are grounded in virtual reality, have emerged as a novel teaching method. Their defining feature is the ability to replicate a realistic tactile sensation in a virtual environment, enabling students to “feel” the procedure in a highly authentic way [14]. One such VS, the Simodont^®^ Dental Trainer (Nissin Dental Products INC., Kyoto, Japan), was developed by Moog (Nieuw-Vennep, The Netherlands) to facilitate dental preparation training in a virtual, representative format. Recent studies suggest that its use in preclinical education effectively enhances students’ manual skills [11,15,16]. The Simodont^®^ allows students to manage and resolve simulated clinical scenarios, mirrors real-life dental practices, tracks student progress, evaluates performance with feedback, and records both the number of attempts and the time required to complete tasks. While both traditional methods and VSs involve an initial financial investment, the VS eliminates material consumption, reduces the need for high teacher-to-student ratios, and provides automated feedback, making it a cost-effective and scalable solution for simulation-assisted learning [3,5,9,17,18]. In pediatric dentistry, the incorporation of VS has yielded promising outcomes, with 88% of students acknowledging that performing dental preparations using Simodont^®^ serves as a valuable complement to traditional preclinical training [9].

For these reasons, we sought to implement the use of virtual simulators (VSs), specifically the Simodont^®^, in the preclinical practices of the dentistry degree. It is crucial that students develop advanced manual skills and abilities before treating pediatric patients, who often pose additional challenges due to their age, such as behavioral management difficulties. The main aim of this project was to incorporate VSs into the preclinical training for the Pediatric Dentistry II module, a 4th-year subject of the Complutense University Madrid (UCM) Dentistry Degree. The specific objectives were (1) to compare the grades achieved in preclinical practice performed on acrylic primary teeth with those obtained using the Simodont^®^; (2) to evaluate whether students’ prior theoretical knowledge influenced their performance in the preclinical practicals; and (3) to gather student feedback regarding their experiences with both training methods.

## 2. Materials and Methods

### 2.1. Study Design and Ethics Statement

An observational, descriptive, and cross-sectional investigation was designed. It was approved as an Educational Innovation Project by the Vicerectorate of Quality of the UCM (Project Number: 340) and received a favorable report from the Ethics Committee of the San Carlos Clinical Hospital (Internal Code: 22/154-E, date of approval: 8 August 2022). The study was conducted by university professors teaching the Pediatric Dentistry II subject to 4th-year students at the Faculty of Dentistry, UCM.

### 2.2. Study Sample and Participants

Firstly, all students enrolled in the subject were informed about this study’s methodology and assured that their participation was voluntary, with the right to withdraw at any time without any consequences. All students who wished to participate voluntarily were included, while those who had previously been enrolled in the Pediatric Dentistry II subject and/or those who did not sign the informed consent for participation were excluded. Additionally, students with language barriers (e.g., Erasmus students) or those who dropped out of the subject were excluded from this study.

### 2.3. Study Procedure and Outcomes

In the first session, a survey was conducted to assess the students’ prior knowledge of class II cavities and pulpotomies in primary dentition (Appendix A). In subsequent sessions, the students performed a class II cavity preparation on an 8.4 tooth and a pulpotomy on an 8.5 tooth, both in the VS and on acrylic temporary teeth in phantoms, which we will refer to as a simulation on acrylic teeth (SAT) (AK-6/PUW, Frasaco GmbH, Tettnang, Alemania) on a pediatric typodont (AK-6/2Z, Frasaco GmbH, Tettnang, Alemania) Once all the dental preparations were completed, the students were asked to complete a modified opinion survey based on Zafar et al. (Appendix B) [9].

For the evaluation of the dental preparations, the rubrics shown in Table 1; Table 2 were created. To avoid evaluation bias, they were assessed by the same professor (Dr. Caleya), who was the principal investigator. To evaluate the treatments performed in the VS, screenshots at maximum resolution from different angles of the cavity preparations were saved in the student’s Simodont^®^ account. In contrast, the preparations performed on the SAT were submitted by the students in anonymized bags. We considered it important to conduct intra- and inter-observer agreement assessments to ensure the validity of the measurements and the reproducibility of this study. Therefore, to analyze both intra- and inter-observer agreement, the principal investigator and a second investigator (Dr. Martin-Vacas) re-scored a random sample of 20 teeth.

### 2.4. Randomization

On the first day, the order of the two learning procedures (VS and SAT) was simply randomized, so that Group A (*n* = 34) and Group B (*n* = 33) were established (Figure 1). The random method was conducted simply with Excel 2021 (Microsoft 365, Microsoft Corporation, Redmond, USA), with the random number generator method.

### 2.5. Statistical Analysis

The normality of the quantitative variables was analyzed with the Kolmogorov–Smirnov and Shapiro–Wilk tests. All tests were performed with the statistical software IBM SPSS Statistics version 24, with a significance level of 95% (*p* ≤ 0.05) and asymptotic or bilateral significance.

Inter- and intra-examiner agreement was obtained with the Kappa coefficient in a random 30% of the sample. The correlation between the prior knowledge test and the scores obtained (Simodont^®^ vs. SAT), the correlation between the scores obtained (Simodont^®^ vs. SAT) in class II cavities, and the correlation between the scores (Simodont^®^ vs. SAT) obtained in pulpotomies were analyzed using the non-parametric tests of Kendall’s Tau and Spearman’s Rho. The differences between the scores of both methods (Simodont^®^ vs. SAT) for each student were evaluated with the Wilcoxon rank test for related samples. In addition, the difference between the components of the rubric (dichotomous) was also analyzed using the McNemar test for related samples. Finally, the student’s perception was analyzed using the chi-squared test.

## 3. Results

### 3.1. Participants

Of the 67 students enrolled in the course, 65 participated. One student declined to participate due to canceling their enrollment in the course, and another was excluded from the study as they were unable to perform the Simodont^®^ treatments.

### 3.2. Influence of Theoretical Knowledge on Preclinical Practices

Previous theoretical knowledge was analyzed, yielding a mean score of 6.806 out of 10 (SD = 1.362). Upon analyzing the distribution of the variable, it was observed that it did not follow a normal distribution (Kolmogorov–Smirnov *p* ≤ 0.001 and Shapiro–Wilk *p* ≤ 0.001).

Regarding the correlation between the prior knowledge test and the score obtained in the dental preparations with Simodont^®^ and the SAT, no statistically significant correlation was found for any of the comparisons (Table 3). In other words, the score on the prior knowledge test did not correlate with the score subsequently obtained in the procedures.

### 3.3. Comparison of Scores Obtained in the Simulated Practices with Simodont^®^ and SAT

With respect to the scores obtained in the simulated Simodont^®^ and SAT, none of the study variables met the normality criteria (Kolmogorov–Smirnov *p* ≤ 0.001 and Shapiro–Wilk *p* = 0.001). For both class II cavities and pulpotomies, the mean score for the SAT method was higher than that obtained with the Simodont^®^ (Table 4).

The correlation of the total scores between both methods was analyzed, revealing positive and significant correlations in the case of class II cavities, but insignificant relationships in the case of pulpotomies. In other words, as the score obtained in the SAT increases, the score obtained in Simodont^®^ also increases, both for class II cavities and pulpotomies. The Wilcoxon rank test for related samples was performed, showing that for both class II cavities and pulpotomies, the scores obtained by the students were significantly higher in the SAT than in Simodont^®^ (Table 5).

The differences in the related scores of the students were analyzed in the five components of the rubric. For class II cavities, it was found that in components 1 (buccal-lingual wall), 2 (amplitude), and 3 (mesial wall) (McNemar *p* > 0.05), there were no statistically significant differences. However, in components 4 (floor) and 5 (angles), the differences were statistically significant (McNemar *p* < 0.001 and *p* = 0.021, respectively). In the case of pulpotomies, significant differences were found in components 2 (amplitude) and 4 (floor) (McNemar *p* = 0.035 and *p* = 0.038, respectively), while the other components of the rubric were statistically insignificant (McNemar *p* > 0.05). In all cases, the score was higher with the SAT than with Simodont^®^.

### 3.4. Assessment of Students’ Perceptions

The results obtained from the perception survey are shown in Table 6. Analyzing questions 1 to 4 regarding the use of Simodont^®^, the data indicated that, significantly, a higher percentage of students agreed that the use of Simodont^®^ helps to improve preclinical skills (41.8%), is useful for training students (52.2%), and facilitates the understanding of clinical procedures (59.7%) in cavity preparations in pediatric dentistry. Additionally, 64.2% of the students believed that Simodont^®^ helps to integrate theoretical knowledge.

When we compared the practical exercises on the SAT with those on Simodont^®^, in questions 6, 8, 10, and 12—concerning the performance of pulpotomies, pulp removal, cavity preparation, and shaping a class II cavity—a high percentage of students, approximately 70%, found these tasks easier to perform on acrylic teeth. No differences were found in questions 9 and 11, which asked whether the creation or shaping of a class II cavity with Simodont^®^ was easy for them.

### 3.5. Intra- and Inter-Examiner Agreement

The intra- and inter-observer agreement values were interpreted following McHugh’s guidelines [19]. All the studied variables demonstrated moderate to almost perfect agreement (Table 7) (Kappa *p* < 0.005 for all values).

## 4. Discussion

Considering the main aim of the present study, we found it valuable to implement the use of Simodont^®^, as authors such as Murbays et al., Zafar et al., and de Boer et al. suggest that it can be a valuable complement in undergraduate dental courses [8,20,21]. Although the scores obtained for Simodont have generally been lower with respect to SAT, we again reiterate that it can be complementary to preclinical practices on acrylic teeth.

Mirghani et al. studied whether there were differences in the acquisition of skills among students in different grades and analyzed their progression as they advanced through the grades. These authors stated that Simodont^®^ is an effective method for measuring students’ performance and improving their training. They observed that, as students gained experience, the time required to complete tasks decreased and accuracy improved, indicating that performance improves with practice [22]. Considering this previous study, as well as our teaching experience, one of the exclusion criteria in our study was to discard repeat students to avoid bias in the results. Another factor measured by Mirghani et al. was the time taken to complete the tasks. However, in our study, we did not measure the time taken by students to perform each tooth preparation, as this was not the focus of our research. We believe it would be interesting for future studies to assess this aspect of performance [22].

In this research, the cavity preparations were performed in 3D, as this is the most accurate simulation of reality. De Boer et al. [21] conducted a study with Simodont^®^ involving 124 students, where the students practiced manual dexterity exercises in both 2D and 3D. They concluded that students preferred the 3D view, and better results were achieved with the three-dimensional vision as it enhanced their perception of the environment.

Although one might assume that a student with a high level of theoretical knowledge would achieve better grades when applying that knowledge to practice, we observed that the theoretical knowledge students possess prior to the pre-clinical practicum does not influence their practicum grades. To the best of our knowledge, we have not found any studies that analyze this specific aspect, so we cannot compare or discuss our results in relation to existing research. We found it interesting to investigate this, because had our results been different, we might have considered requiring a minimum level of theoretical knowledge as a prerequisite for undertaking the practicum.

Due to the digital advancements in simulated teaching in recent years, we found it very interesting to compare the scores obtained in cavity preparations on acrylic primary teeth (SAT), which is the technique traditionally used in preclinical practices, with those obtained using Simodont^®^. We observed that the scores were higher when using acrylic teeth, both for pulpotomy and class II cavity preparations. It is important to note that, although the aim of our study was to implement the use of Simodont^®^ in the preclinical practices of the Pediatric Dentistry II course, students in our faculty have already practiced with virtual simulators in their first, third, and fourth years of study. We believe that these score differences may be due to the fact that fourth-year students are more accustomed to practicing with SAT than with Simodont^®^. Hattori et al. [23] conducted a study similar to ours, where 30 students performed full-cast crown preparations on both Simodont^®^ and SAT on the right first molar in the mandible. These authors also used evaluation rubrics, as we did, and their results aligned with ours: the score for teeth prepared using the conventional technique on acrylic teeth was higher than for teeth prepared using Simodont^®^. They also observed statistically significant differences in some of the items analyzed, which mirrors our findings. Murbays et al. [20] sought to evaluate dental students’ performance by comparing the scores for class I cavity preparations using traditional methods (phantoms and typodonts with acrylic teeth) and virtual simulation (VS). They found that the use of VS significantly improved student performance. However, authors such as Vincents found very similar results when performing class II cavities with Simodont^®^ and SAT. Although the scores were higher for acrylic teeth in our study, as well as in Hattori et al. and Murbays et al., Plesas et al. [18,20] suggested combining and alternating traditional simulation methods with virtual simulation methods. However, there is insufficient evidence to recommend one method over the other.

Regarding students’ opinions on the use of Simodont^®^, we found studies such as the one conducted by Barkr [15] in 2011, where 4th- and 5th-year students used Simodont^®^ and were asked to evaluate various aspects by filling out pre- and post-experimental questionnaires. They concluded that these simulators should be used in conjunction with other conventional educational methods. A noteworthy study by Zafar [9] involved students performing a pulpotomy and crown carving during preclinical pediatric dentistry practicals, after which they completed a previously validated survey, modified by the authors. Analyzing our results, we find that they align with the findings of these previous authors, as we observed that although Simodont^®^ helps to improve preclinical skills, facilitates training for dental preparations, and aids in understanding clinical procedures, when comparing Simodont^®^ with traditional techniques, students still prefer performing preclinical practices on acrylic teeth.

This research has some limitations. On the one hand, regarding the previous experience with the compared methods, it is important to note that the participants in our study are 4th-year students who have already completed preclinical practice on acrylic teeth in previous years, as well as preclinical practice on Simodont^®^, although the latter received fewer training hours. Limited experience with the simulation method may have been a source of bias, as it has been shown that students achieve improvement in manual skills as they gain experience [13]. Furthermore, the skills acquired through the conventional method and the digital method are difficult to compare, as they are not exclusively motor skills. On the other hand, we would have liked to create more subgroups to compare gender and skills, but the limited sample size did not allow for this, making it impossible to control some of the variables inherent to the students.

Therefore, we believe it is important to continue conducting further studies to explore this area. Therefore, we believe it would be interesting for future studies to consider implementing a crossover design to address this limitation.

## 5. Conclusions

The satisfaction of students using Simodont^®^ in the preclinical practices of the Pediatric Dentistry II subject is inconclusive, with ambiguous results. However, the scores achieved in dental preparations on acrylic teeth are higher than those performed with Simodont^®^. The students’ opinions on performing tooth preparations on primary teeth with Simodont^®^ were generally positive. However, when compared to the traditional method, students still preferred the latter. Therefore, Simodont^®^ can serve as a complementary tool to the classical learning technique, but it is not a substitute for it.

## Figures and Tables

**Figure 1 dentistry-13-00051-f001:**
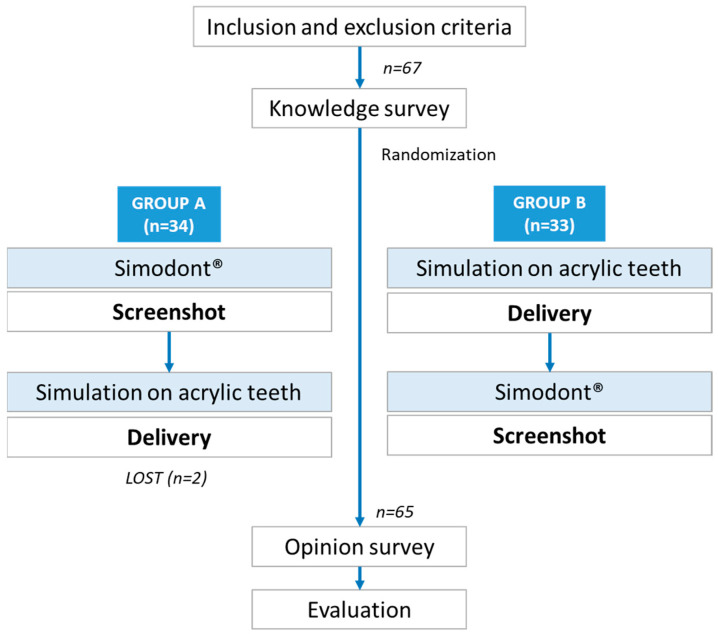
Study procedure.

**Table 1 dentistry-13-00051-t001:** Class II interproximal cavity evaluation rubric.

	Correct (1 Point)	Incorrect (0 Points)
1. Vestibular and lingual wall converging towards the occlusal surface and outside the contact surface of the adjacent tooth.	Yes	No
2. Cavity width no greater than 1.5 mm bucco-lingually (approximately 1/3 of the intercuspidal distance).	Yes	No
3. Parallel or slightly divergent mesial or distal cavity wall	Yes	No
4. Flat floor	Yes	No
5. Smooth and rounded angles	Yes	No

**Table 2 dentistry-13-00051-t002:** Pulpotomy cavity evaluation rubric.

	Correct (1 Point)	Incorrect (0 Points)
Complete removal of the pulp chamber roof	Yes	No
2. Weakness of the vestibular and/or lingual walls	No	Yes
3. Weakness of mesial and/or distal walls	No	Yes
4. Complete removal of pulp chamber tissue	Yes	No
5. Depth of the cavity > pulp chamber	No	Yes

**Table 3 dentistry-13-00051-t003:** Influence of theoretical knowledge scores on preclinical practices.

Previous Knowledge	Kendall Tau		Spearman Rho	
	Correlation Coefficient	*p*-Value	Correlation Coefficient	*p*-Value
Class II Simodont^®^	−0.007	0.948	−0.006	0.959
Class II SAT	0.130	0.198	0.177	0.156
Pulpotomy Simodont^®^	−0.014	0.891	−0.017	0.890
Pulpotomy SAT	−0.114	0.263	−0.139	0.265

SAT. Simulation on Acrylic Teeth.

**Table 4 dentistry-13-00051-t004:** Results of the rubrics of the treatments evaluated.

		Mean	95% CI		SD
			Lower Limit	Upper Limit	
Class II	Simodont^®^	1.9692	1.9889	2.2496	1.13150
	SAT	2.5692	2.2627	2.8758	1.23705
Pulpotomy	Simodont^®^	2.9231	2.6984	3.1477	0.90671
	SAT	3.6000	3.3237	3.8763	1.11524

CI. Confidence Interval. SD. Standard deviation. SAT. Simulation on Acrylic Teeth.

**Table 5 dentistry-13-00051-t005:** Relationships obtained between total class II scores and pulpotomy scores with Simodont^®^ or SAT.

Simodont^®^ vs. Simulation on Acrylic Teeth	Kendall Tau		Spearman Rho		Wilcoxon Rank Test
	Correlation Coefficient	*p*-Value	Correlation Coefficient	*p*-Value	*p*-Value
Class II	0.289	0.005 *	0.343	0.005 *	0.001 *
Pulpotomy	0.103	0.331	0.123	0.330	<0.001 *

* Statistical Significance (*p* ≤ 0.05).

**Table 6 dentistry-13-00051-t006:** Student’s perception survey.

		Agree		Neutral		Disagree		ꭓ^2^
		*n*	%	*n*	%	*n*	%	*p*-Value
Simodont^®^ Perception	1	28	41.8	26	38.8	11	16.4	0.019 *
	2	35	52.2	21	31.3	9	13.4	<0.001 *
	3	40	59.7	20	29.9	5	7.5	<0.001 *
	4	43	64.2	20	29.9	2	3	<0.001 *
Simodont^®^ vs. SAT	5	5	7.5	23	34.3	37	55.2	<0.001 *
	6	49	73.1	8	11.9	8	11.9	<0.001 *
	7	14	20.9	19	28.4	32	47.8	0.019 *
	8	46	68.7	16	23.9	3	4.5	<0.001 *
	9	21	31.3	25	37.3	19	28.4	0.650
	10	47	70.1	13	19.4	5	7.5	<0.001 *
	11	20	29.9	21	31.3	24	35.8	0.819
	12	47	70.1	14	20.9	4	6	<0.001 *

* Statistical Significance (*p* ≤ 0.05).

**Table 7 dentistry-13-00051-t007:** Inter-examiner and Intra-examiner agreement in dental preparation evaluations with rubrics.

	Inter-Examiner				Intra-Examiner			
	Simodont^®^		SAT		Simodont^®^		SAT	
	Kappa Value	Agreement	Kappa Value	Agreement	Kappa Value	Agreement	Kappa Value	Agreement
**Question**	**Class II**							
1	0.894	Strong	1	Almost perfect	1	Almost perfect	0.900	Strong
2	0.792	Moderate	0.700	Moderate	0.898	Strong	0.900	Strong
3	0.681	Moderate	0.898	Strong	0.886	Strong	0.700	Moderate
4	0.828	Strong	1	Almost perfect	0.608	Moderate	0.659	Moderate
5	0.681	Moderate	0.794	Moderate	0.898	Strong	1.000	Almost perfect
**Question**	**Pulpotomy**							
1	0.765	Moderate	0.692	Moderate	1	Almost perfect	0.886	Strong
2	0.612	Moderate	1	Almost perfect	1	Almost perfect	1	Almost perfect
3	0.737	Moderate	0.773	Moderate	1	Almost perfect	1	Almost perfect
4	1	Almost perfect	0.643	Moderate	0.894	Strong	0.773	Moderate
5	0.828	Strong	0.794	Moderate	1	Almost perfect	0.900	Minimal

SAT. Simulation on Acrylic Teeth.

## Data Availability

The raw data presented in this study are available on request from the corresponding author.

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
