# Peer review of "Implementation of Virtual Reality in Preclinical Pediatric Dentistry Learning: A Comparison Between Simodont® and Conventional Methods"

_dentistry, 2025, doi:10.3390/dj13020051_

Round 1
Reviewer 1 Report
Comments and Suggestions for Authors
Thank you for your paper “Implementation of Virtual Reality in preclinical Pediatric Dentistry learning: a comparison between Simodont® and conventional methods”. The submission addresses an innovation in pediatric dentistry education.
The article is well written and really interesting. The manuscript is clear and well-structured. All the cited references are recent publications and relevant. There was sufficient number of participants of the study. All important information are given in the material and methods section. Is easy to understand the figures.
Author Response
Thank you for your comments and for your valuable time in reviewing our work.
Best regards.
Reviewer 2 Report
Comments and Suggestions for Authors
Dear Authors,
All my comments are attached to the PDF file.
Kind regards and good luck

Author Response
Thank you for your comments and for your valuable time. Attached are the responses to your comments. Thank you. Best regards.

Reviewer 3 Report
Comments and Suggestions for Authors
This manuscript entitled Implementation of Virtual Reality in Preclinical Pediatric Dentistry Learning: A Comparison Between Simodont® and Conventional Methods is very interesting and could be of significant importance in the future due to the implementation of new methods in the learning process.
Although I find this article valuable, I would like to suggest some changes:
-
Table 6 and Figure 4 present the same information, so one of them should be deleted. I suggest removing Figure 4.
-
It is unclear why the authors used Kendall’s Tau and Spearman’s correlation in Table 5. Could the authors clarify the rationale for using both methods?
-
Please add the limitations of the study.
-
In my opinion, Figure 1 and Figure 2 are unnecessary. Please consider removing them, or placing them at the end of the article as supplementary files.
Please change the reference citation according to the Dentistry guidelines
Author Response
Thank you for your comments and for your valuable time. Attached are the responses to your comments. Attached are the responses to your comments.
Thank you. Best regards.

Round 2
Reviewer 2 Report
Comments and Suggestions for Authors
Dear Authors,
Thank you for the feedback.
This is a revised version of your original manuscript. You have corrected it accordingly and it can be accepted without any structural changes now.
Best regards and good luck
Reviewer 3 Report
Comments and Suggestions for Authors
Thank you for accepting the suggestions. In my opinion, the article is now significantly improved.